# Effect of Loblolly Pine (*Pinus taeda* L.) Hemicellulose Structure on the Properties of Hemicellulose-Polyvinyl Alcohol Composite Film

**DOI:** 10.3390/molecules28010046

**Published:** 2022-12-21

**Authors:** Huaizhi Pan, Biao Zheng, Hui Yang, Yingying Guan, Liuyang Zhang, Xiaoli Xu, Aimin Wu, Huiling Li

**Affiliations:** 1State Key Laboratory for Conservation and Utilization of Subtropical Agrobioresources, South China Agricultural University, Guangzhou 510642, China; 2Guangdong Key Laboratory for Innovative Development and Utilization of Forest Plant Germplasm, College of Forestry and Landscape Architecture, South China Agricultural University, Guangzhou 510642, China; 3Instrumental Analysis and Research Center, South China Agricultural University, Guangzhou 510642, China

**Keywords:** loblolly pine, hemicellulose, graded ethanol precipitation, hemicellulose-polyvinyl alcohol composite film

## Abstract

Hemicellulose is the second most abundant natural polysaccharide and a promising feedstock for biomaterial synthesis. In the present study, the hemicellulose of loblolly pine was obtained by the alkali extraction-graded ethanol precipitation technique, and the hemicellulose-polyvinyl alcohol (hemicellulose-PVA) composite film was prepared by film casting from water. Results showed that hemicellulose with a low degree of substitution is prone to self-aggregation during film formation, while hemicellulose with high branching has better compatibility with PVA and is easier to form a homogeneous composite film. In addition, the higher molecular weight of hemicellulose facilitates the preparation of hemicellulose-PVA composite film with better mechanical properties. More residual lignin in hemicellulose results in the better UV shielding ability of the composite film. This study provides essential support for the efficient and rational utilization of hemicellulose.

## 1. Introduction

With the increasing consumption of petroleum resources, the search for renewable alternatives has become more and more urgent. Hemicellulose, one of the main components of lignocellulose, is the second largest biomass component after cellulose. It is one of the most reliable resources which has the advantages of extensiveness, reproducibility, etc. [1]. The highly biocompatible and degradable polymer materials prepared with hemicellulose as raw materials have the potential to replace traditional petroleum-based materials.

Hemicellulose is a group of heteropolysaccharides in plant cell walls which are chemically bonded to cellulose and lignin. Hemicellulose has a complex structure and is composed of a variety of glycans [2]. Different raw materials have different components. The glycosyl units of hemicellulose mainly include D-xylose, D-mannose, L-arabinose, D-glucose, D-galactose, D-glucuronic acid, and D-galacturonic acid [3]. The hemicellulose in broadleaved trees and grass species is mainly xylan, which has β-(1-4)-D-xylopyranosyl (Xyl*p*) and a large number of substituents in the side chain [4]. The substituents mainly include α-L-arabinofuranose (Ara*f*), α-D-glucuronic acid (GlcA), or/and 4-O-methyl-α-D-glucuronic acid (MeGlcA). Xylans are usually modified with acetyl groups, especially for dicot plants which has a higher degree of acetylation [5,6]. In coniferous wood, the main structure of hemicellulose is mannan. The main chain of mannan is composed of β-(1-4)-D-mannopyranosyl (Man*p*) and β-(1-4)-D-glucopyranosyl (Glc*p*), with a small amount of α-D- galactopyranosyl (Gal*p*) and acetyl groups as side chains [7]. The degree of acetylation of softwood hemicellulose is lower than that of broadleaf wood and grass. Among them, mannan mainly exists in two ways; one is galactoglucomannan, and the other is glucomannan, both of which are composed of galactose linked by mannose by α-(1-6) glycosidic bonds [8].

In recent years, biodegradable or edible films prepared from hemicellulose have attracted much attention as alternatives to petroleum-based films. The properties of the membrane are closely related to the physicochemical properties of raw materials [9,10]. Most research focused on xylan membranes. Research showed that the side chain has influence on the mechanical properties of the membrane. The appropriate degree of substitution can improve the mechanical properties of the membrane [11]. In addition, side chains also affect the crystallinity of the hemicellulose composite films in the process of film formation [12]. Hemicellulose with a high degree of substitution is amorphous, while hemicellulose with a low degree of substitution has obvious crystalline structure. The crystalline structure of hemicellulose will reduce mechanical properties and enhance the thermal stability of membrane. Moreover, the solubility of hemicellulose was demonstrated to be influenced by its degree of polymerization (DP), thus affecting the characteristic of final products [13]. The larger the degree of polymerization of the hemicellulose is, the lower the solubility it has. Membrane prepared with high DP hemicellulose is easy to be broken in the drying process [14]. Furthermore, lignin in hemicellulose can act as a reinforcer to accelerate the formation of films [15]. Therefore, it is essential to understand the relationship between the molecular structure of mannan and the physicochemical properties of mannan films.

The preparation of hemicellulose films usually requires the addition of plasticizers such as polyvinyl alcohol, sorbitol, glycerin, etc. These plasticizers can facilitate film formation. PVA is a common plasticizer, which is conducive to the formation of the film [16,17]. Xu studied the effects of different additives (glycerol, xylitol and sorbitol) on the mechanical properties of hemicellulose composite membranes. Results showed that glycerol could greatly enhance the tensile resistance of the film [18]. Hartman found that the addition of sorbitol can increase the oxygen resistance of galactomannan film [19]. Chen demonstrated that adding PVA and nano chitin was beneficial to improve the tensile properties, oxygen resistance, and light transmittance of the film [20]. Therefore, suitable additives may effectively improve the performance of hemicellulose film.

Loblolly pine is widely distributed in China, and a large amount of waste is generated in the process of wood processing [21]. In this study, loblolly pine hemicellulose was prepared by sodium hypochlorite delignification-alkali extraction-graded ethanol precipitation. In addition, loblolly pine hemicellulose membrane was prepared by film casting from water using PVA as plasticizer. The physicochemical properties of pine hemicelluloses precipitated with different concentration of ethanol as well as the synthesized hemicellulose film were analyzed. Furthermore, the influence of the structure of hemicellulose on the properties of the synthesized hemicelluloses film was discussed.

## 2. Results and Discussion

### 2.1. Yields of Hemicellulosic Fractions Precipitated by Different Ethanol Concentrations

In this study, lignin was first removed by sodium chlorite, and then hemicellulose in the solid residues was extracted with 20% KOH solution (adding 2% H_3_BO_3_) [22]. The dissolved hemicellulose in the lye was further precipitated by gradually increasing the concentration of ethanol from 30% to 75%. The yields of the precipitated alkali-soluble hemicellulosic fractions are presented in Table 1. The yields of hemicellulose decreased from 8.86% to 1.07% with an increase of the ethanol concentration from 30% to 75%. The total amount of the precipitated hemicellulose was 19.02%, which corresponded to 85.63% of original hemicellulose. This suggests that the vast majority of hemicellulose can be separated at an ethanol concentration below 75%, which is consistent with the results in the literature [23,24].

### 2.2. Chemical Composition of Alkali-Soluble Hemicellulosic Fractions

The contents of neutral sugar composition and lignin of alkali-soluble hemicellulose precipitated by different ethanol concentrations are shown in Table 2. The dominating sugars of the hemicellulosic fractions are identified as mannose (46.58–51.91%), followed by xylose (14.9–23.97%), glucose (9.19–15.81%), galactose (7.66–11.02%) and arabinose (2.97–3.57%). It was also found that lignin existed in all hemicellulose, which is due to the various chemical bonds between hemicellulose and lignin. Even though the material was delignified with sodium chlorite and glacial acetic acid, the lignin cannot be completely removed, resulting in a small amount of lignin remaining in the extracted hemicellulose. The lignin content in hemicellulose increased with increasing ethanol concentration, indicating that more lignin remained in the precipitated hemicellulose under higher ethanol concentration. [25]. The content of xylose and glucose decreased with increasing ethanol concentration, while the content of mannose increased, indicating that low concentration of ethanol is favored for the precipitation of xylan and high concentration of ethanol is favored for the precipitation of mannan. The solubility of hemicellulose is related to the degree of substitution. The more side chains or branches it owns, the better solubility could be observed [26]. As seen from Table 2, DS1 increased from 0.14 to 0.21 and DS2 increased from 0.12 to 0.17 as the ethanol concentration increased from 30% to 60%. When the ethanol concentration was increased to 75%, both DS1 and DS2 decreased slightly, but still had higher DS values than that of H30 and H45. The result implied that lower ethanol concentrations favor the precipitation of linear hemicellulose, while higher ethanol concentrations was favored to generate highly substituted multi-branched hemicellulose.

### 2.3. Molecular Weight Analysis

The weight-average molecular weights (*Mw*) and number-average molecular weights (*Mn*) as well as the polydispersity index (*Mw*/*Mn*) of the hemicellulosic fractions are shown in Table 3. *Mw* of the extracted hemicellulose increased slightly from 71,295 g/mol to 72,619 g/mol when the ethanol concentration is increased from 30% to 45%. While the ethanol concentration increases from 45% to 75%, *Mw* decreases to 25,182 g/mol. The trend of *Mn* is basically the same as that of *Mw*. Based on the results of molecular weights and the chemical composition analysis, hemicelluloses with larger molecular weight and longer main chain are easier to be precipitated by lower ethanol concentrations (30% and 45%), while hemicelluloses with shorter main chain, highly branched and smaller molecular weight are favored to be precipitated by the higher ethanol concentration (60% and 75%). However, since the lignin content in hemicellulose increases with the increment of ethanol concentration. The higher Mw/Mn value of hemicellulose at the higher ethanol concentration may be from the higher content of lignin. In addition, all hemicellulosic fractions showed relatively low Mw/Mn in the range of 1.46 to 1.88, which implied that the hemicellulose is chemically and structurally homogeneous under the given conditions [23]. Therefore, it could be concluded that the hemicellulose precipitated by the higher concentration of ethanol has a relatively wide molecular weight distribution, while the hemicellulose precipitated by 45% ethanol has the narrowest molecular weight distribution, indicating that the degree of polymerization of H45 has the best uniformity.

### 2.4. FT-IR Spectra Analysis

The FTIR spectra of the four hemicellulosic fractions extracted by different ethanol concentrations are shown in Figure 1. The absorption peak at 1630 cm^−1^ corresponds to the vibrational absorption of the C-Ph bond on lignin [27]. The intensity of this peak increased with increasing ethanol concentration, suggesting that the amount of lignin residues in the extracted hemicellulose increases with increasing ethanol concentration. The strong absorption peak at 1415 cm^−1^ corresponds to the vibrational bending of C-H or O-H on the hemicellulose sugar ring [28], the absorption peaks at 1326 cm^−1^ and 1047 cm^−1^ are originated from the vibrations of C-C and C-O skeletons, respectively [29,30]. 1258 cm^−1^ is ascribed from the O-H or C-O bending vibrations [31]. The shoulder peak at 1162 cm^−1^ is the characteristic absorption peak of the arabinosyl side chain [32], indicating that xylan with arabinose side chains could be precipitated by different concentrations of ethanol. 873 cm^−1^ and 798 cm^−1^ are the characteristic absorption peaks of galactose residues and mannose residues, respectively, indicating that ethanol-precipitated hemicellulose has mannose backbone and galactose side chains [33]. The intensity of the absorption peak at 798 cm^−1^ increased with the increasing of ethanol concentration, indicating that the high concentration of ethanol facilitated the precipitation of mannans, which was consistent with the results of hemicellulose composition analysis.

### 2.5. HSQC NMR Analysis

To further elucidate the structural information of the hemicellulosic polymers, H45 was selected for the HSQC NMR analysis, and the results are shown in Figure 2. Attribution of HSQC NMR spectrum were assigned according to the literatures [34,35,36,37,38,39,40]. Signals at 99.74/4.59 (C1-H1), 70.07/3.92 (C2-H2), 70.19/3.85 (C3-H3), 75.63/3.66 (C4-H4), 76.13/3.30 (C5-H5) and 60.16/3.79 (C6-H6) ppm were correlated with (1→4)-β-D-Man*p* units, 70.07/3.92 (C3-H3) and 64.92/3.83 (C6-H6) ppm are the cross-signal peaks of (1→4,6)-β-D-Man*p* units. Five cross-signal peaks appearing at 101.95/4.36 (C1-H1), 73.69/3.21 (C2-H2), 72.34/3.43 (C3-H3), 75.25/3.50 (C5-H5) and 59.86/3.62 (C6-H6) ppm were associated with (1→4)-β-D-Glc*p* units. 69.01/3.50 (C2-H2), 70.01/3.67 (C3-H3), 67.37/3.82 (C4-H4), 72.56/3.63 (C5-H5), 57.80/3.61 (C6-H6) ppm are the five crossover signals arising from α-D-Gal*p* units. These cross-peak signals indicating the presence of galactoglucomannan in H45 hemicellulose. Six major signals appeared at 101.44/4.30 (C1-H1), 72.62/3.19 (C2-H2), 75.28/3.37 (C3-H3), 75.95/3.76 (C4-H4), 65.05/3.17 (C5ax-H5ax) and 62.71/3.91 (C5eq-H5eq) ppm are associated with (1→4)-β-D-Xyl*p* units. Five Ara*f*-related signals appear at 106.75/5.27 (C1-H1), 80.24/4.02 (C2-H2), 75.84/3.80 (C3-H3), 84.64/4.11 (C4-H4) and 62.51/3.78 (C5-H5) ppm. Five signals of 4-O-Me-α-D-Glc*p*A units appeared at 93.44/5.06 (C1-H1), 70.60/3.44 (C2-H2), 70.72/3.75 (C3-H3), 81.87/3.10 (C4-H4) and 71.69/4.22 (C5-H5) ppm, where the OCH_3_ (3.34/59.60 ppm) signal is generated by methylation of glucuronide on the side chain of the xylose residue. Signals associated with (1→4)-β-D-Xyl*p*-2-O-Glc*p*A units appear at 96.05/4.44 (C1-H1), 62.51/3.97 (C5eq-H5eq) and 62.56/3.24 (C5ax-H5ax) ppm.

Therefore, combined with the results of sugar analysis, FT-IR and NMR analysis, it could be concluded that the H45 were mainly composed of galactoglucomannan (GGM) and 4-*O*-methylglucuronoarabinoxylan (GAX).

Note: In HSQC NMR spectra, M, (1→4)-β-D-Man*p*; M′, (1→4,6)-β-D-Man*p*; G, (1→4)-β-D-Glc*p*; Gal, α-D-Gal*p* units; X, (1→4)-β-D-Xyl*p*; A, α-L-Ara*f* units; U, 4-O-Me-α-D-Glc*p*A; XU, (1→4)-β-D-Xyl*p*-2-O-(4-O-Me-α-D-Glc*p*A); ax, axial; eq, equatorial.

### 2.6. Thermal Stability Analysis of Hemicellulose

The thermal stability of the four hemicellulosic fractions precipitated by different concentrations of ethanol were analyzed by thermogravimetry (TG), and the results are shown in Figure 3. The thermal degradation of hemicellulose is divided into three stages. The first stage, before 120 °C, is due to the evaporation of small amount of water contained in the hemicellulose. The second stage, during 120–360 °C, is ascribed to the pyrolysis of hemicellulose to produce volatiles. The third stage, after 360 °C, is the stage of being completely pyrolyzed [41].

From the DTG curve analysis, the differences in the thermal stability of the four hemicellulose were mainly in the second stage, where H60 and H75 reached maximum mass loss rates of 0.97%/°C and 0.68%/°C at 217 °C and 225 °C, while H30 and H45 reached maximum mass loss rates of 0.41%/°C and 0.58%/°C at 256 °C and 267 °C. This variation may be due to the difference in hemicellulose molecular weight and substitution degrees. According to the literature, the higher the molecular weight of hemicellulose is, the better thermal stability could be obtained, and the side chains are more susceptible to pyrolysis than the main chain [29,42]. In addition, hexoses are much more thermally stable than pentoses. The molecular weights of H75 and H60 are much lower than those of H30 and H45, and H75 and H60 have more side chains. This indicates that the hemicelluloses obtained from lower ethanol concentration have better thermal stability than those obtained from higher ethanol concentration. The high content of the solid residues in H30 and H45 after 300 °C are biochar formed through the pyrolysis of the material [43]. The results showed that the thermal stability of H30 and H45 fractions was higher than that of H60 and H75 fractions, which was depended on many factors, such as sugar types, molecular weights and branching degree of hemicelluloses [23].

### 2.7. Morphological Observation of Composite Film

The hemicellulose-polyvinyl alcohol composite film was prepared by film casting from water. Figure 4 shows the photographs and SEM images of the composite films. It can be seen from Figure 4A that all hemicellulose-PVA composite films show an intact, continuous and smooth appearance. The H0-PVA composite film is colorless and transparent. However, the composite films with added hemicellulose were slightly yellow, which was caused by a small amount of lignin remaining in the hemicellulose. The surface micromorphology of the composite film was observed by SEM and the images are shown in Figure 4B. It is observed from the figure that the surface of the H0-PVA composite film without added hemicellulose has no fragmentation and pores and is smooth without the presence of tiny bumpy bodies. However, there are many micro-convenes on the surface of H30-PVA composite film. The surface of H45-PVA is smooth without the presence of micro-convexity, but there are many small round holes. Compared with H30 and H60, H45 has more side chains, and the side chain groups will act as spatial blockers to prevent hemicellulose from self-aggregating, thus increasing the hydrogen bonding between hemicellulose and PVA or water molecules, thereby increasing the solubility [44].

### 2.8. Mechanical Properties of Composite Film

The mechanical stress and strain of the composite film is an important indication of rigidity and toughness. In the present study, the mechanical stress and strain of hemicellulose-PVA composite films were analyzed by material tension tester, and the results are shown in Figure 5. Among all samples, H0-PVA has the lowest maximum tensile strength. The maximum tensile strength of the composite films with added hemicellulose was higher than that of the pure PVA composite film (H0-PVA), indicating that hemicellulose was beneficial to enhance the tensile strength of the PVA composite film. H45-PVA composite film showed the best tensile strength (30 Mpa) and tensile strain (7%), which may be ascribed to the reason that H45 hemicellulose has the higher molecular weight and the larger number of side chains, which resulted in a higher hydrogen bonding force between PVA molecules. There are many small pores on the surface of H45-PVA composite film, and these small pores may play a buffering role in the stress process to enhance the tensile stress of the composite film. There are also many side chains in H60 hemicellulose, but the maximum tensile strength of H60-PVA composite film is only 24 MPa. This phenomenon may be due to the reason that the H60 contains more lignin, and the self-aggregation and lack of uniform dispersion of lignin can cause uneven stress transfer, which leads to a decrease in the mechanical properties of the composite film. [45]. The maximum tensile strength of H30-PVA film was 27 MPa, but its maximum tensile strain was only 3%. Dispersion of hemicellulose possibly acted as reinforcement, improving strength of the polymer matrices which diversely reduced flexibility of the films [46]. According to the Chinese food packaging film material standard GB 30768, the maximum tensile strength should be no less than 30 MPa and the maximum tensile strain should be equal to or higher than 5% [17]. Therefore, H45-PVA composite film meets the mechanical requirements of packaging materials, which lays the foundation for research on the application of hemicellulose composite film in food packaging. These natural based materials have high potential to serve as sustainable polymeric packaging which reduced the environmental impact of petroleum-based plastic [47,48].

### 2.9. Transparency of Composite Film

In the present study, the transparency performances of the hemicellulose-PVA composite films were analyzed using UV-Vis spectrophotometer, and the results are shown in Figure 6. Compared with pure PVA film (H0-PVA), hemicellulose-PVA composite film presented the better light shielding ability. Among all samples, H60-PVA composite film has the best UV-shielding ability. This is due to the highest amount of lignin residue in H60 hemicellulose [49]. The transmission curves of films fluctuated at 280 nm, which may be due to the existence of the amino groups [50]. In the visible areas, the transparency was in the order of H60-PVA > H45-PVA > H30-PVA. The more hemicellulose side chains interact with PVA can reduce the distance between polymers and increase the denseness of the film, resulting in a certain shielding effect on visible light [18]

### 2.10. Thermal Stability of Composite Films

The effect of hemicellulose structure on the thermal stability of the composite film was further investigated by thermogravimetric analysis (Figure 7). It can be seen from the Figure 7 that there are four pyrolysis stages of the composite films. The first stage was before 200 °C, the second stage was 200–360 °C, the third stage was 360–480 °C, and the fourth stage was after 480 °C. The first stage is due to the volatilization of water in the samples. The second stage is the rapid loss of polymer mass. The starting pyrolysis temperatures of all composite films are at 200 °C. The temperature at maximum mass loss of hemicellulose-PVA composite films is lower than that of pure PVA film (H0-PVA), and the maximum pyrolysis rates of H45-PVA and H60-PVA are much higher than that of H0-PVA, indicating that the poorer thermal stability of hemicellulose leads to the poorer thermal stability of the composite films. The third stage can only be observed on the DTG curve of composite film (Figure 7) but not on the DTG curve of hemicellulose (Figure 3), indicating that this stage is caused by the pyrolysis of PVA instead of the hemicellulose pyrolysis. The fourth stage is the charring process. The mass of the final residues of hemicellulose-PVA composite film was much higher than that of pure PVA film (H0-PVA), indicating that hemicellulose generates more char residue in nitrogen atmosphere [51].

## 3. Materials and Methods

### 3.1. Materials

Four-year-old loblolly pine about 3 m high was obtained from National Loblolly Pine Seed Base on Yingde City, Guangdong Province, China. Before experiments, about 30 cm wood section was taken from the base of loblolly pine (about 1 m from the ground). Samples were decorticated, ground and sieved to 40–60 mesh size. The obtained particles were dewaxed with acetone/ethanol (2:1, *v*/*v*) in a Soxhlet apparatus at 90 °C for 8 h, and then oven-dried at 60 °C to constant weight [39,52]. Chemical composition of loblolly pine material after dewaxed was determined in accordance with the analysis procedure of National Renewable Energy Laboratory (NREL/TP-510-42618), and the contents of cellulose, hemicellulose and lignin were 43.86%, 22.21% and 33.93%, respectively. Standard reagents were HPLC grade and were purchased from Sigma-Aldrich. PVA (Mw ~145000) was purchased from Aladdin-Reagent (Shanghai, China). Other chemicals used in the experiment were of analytical grade and were obtained from Shanghai Macklin Reagent Co. (Shanghai, China).

### 3.2. Extraction and Fractionation of Hemicellulose

Schematic illustration for the fractionation of alkali-soluble hemicellulose from loblolly pine is illustrated in Figure 8. In brief, 2 g of dewaxed material was added to 65 mL of distilled water, followed by the addition of 0.5 mL of glacial acetic acid and 0.6 g of sodium chlorite. The mixture was magnetic stirred for 6 h at 75 °C. 0.5 mL of glacial acetic acid and 0.6 g of sodium chlorite were added every hour until the color of raw materials became white, then dried to obtain holocellulose [26]. About 10 g holocellulose was extracted with 20% KOH solution (adding 2% H_3_BO_3_) with the solid-to-liquid ratio of 1:20 (g/mL) at 25 °C for 10 h [53]. After the reaction, the filtrate was adjusted to pH 5.5 with acetic acid and then concentrated under reduced pressure to 30 mL. Then, sequentially, fractionated by graded precipitations at different ethanol concentrations of 30%, 45%, 60%, and 75% (*v*/*v*), respectively. After centrifugation, each fraction of precipitate was washed with 70% ethanol. The obtained components were further freeze-dried. The hemicellulose fractions were named H30, H45, H60, and H75, corresponding to the ethanol concentration of 30%, 45%, 60%, and 75%, respectively.

### 3.3. Characteristics of Alkali-Soluble Hemicellulosic Fractions

The monosaccharide composition of hemicellulose was analyzed by high-performance anion-exchange chromatography (HPAEC, Metrohm 940, Switzerland). The sample (25 mg) was hydrolyzed at 108 °C for 2.5 h after adding 0.625 mL of 72% H_2_SO_4_ and 6.75 mL of deionized water. After the reaction, the hydrolysate was filtered using a 0.22 µm filter head. The filtrate was diluted 50 times with deionized water for analysis. The mobile phase was 2 mmol sodium hydroxide and 75 mmol sodium acetate. The column temperature was 30 °C and the flow rate was 0.5 mL/min. The contents of D-glucose, D-mannose, D-xylose, D-galactose and L-arabinose were calculated by external standard method [54]. The significance analysis of the data was performed by Duncan’s multiple range test.

The molecular weights of hemicelluloses were determined by gel permeation chromatography (GPC, Wyatt, America) with an Agilent PL aqua gel-OH MIXED-H column (300 × 7.5 mm) and an evaporative light-scattering detector. Sodium phosphate buffer (5 mM, pH = 7.5) containing NaCl (0.02 N) was employed as the eluent with a flow rate of 0.5 mL/min at 35 °C [55]. Data were presented as the average of two measurements.

FTIR spectroscopy (Vertex 70, Bruker, Germany) was used to analyze the structure of different fractions of hemicellulose with a scan range of 2000–500 cm^−1^ and a resolution of 4 cm^−1^ for 32 scans. The HSQC spectrum of hemicellulose was obtained by Bruker AV600 spectrometer (Bruker, Germany). The specific steps and parameters were as follows: 50 mg of KOH-extracted hemicellulose was dissolved in 500 µL of D_2_O. Over a t_1_ spectral width was 20,000 Hz, t_2_ spectral width was 4000 Hz, and the relaxation time was 1.5 s [56].

A thermogravimetric analyzer (TA Q500, American) was used to test the thermal stability of the hemicellulose. The nitrogen flow rate was 20 mL/min. About 10 mg of hemicellulose was heated from 55 °C to 750 °C at 10 °C/min, and the mass change was recorded [40].

### 3.4. Preparation of Hemicellulose-PVA Composite Film

Hemicellulose-PVA composite film was prepared by film casting from water [12,43]. 0.15 g of hemicellulose and 0.45 g of PVA were added into 8 mL of deionized water. The components were stirred magnetically at 95 °C for 45 min to form a homogeneous suspension, and then poured into a glass Petri dish (60 mm diameter). The mixture was vacuumized immediately after pouring to remove air bubbles. The solution was dried at room temperature for 3–5 days. In addition, PVA films without hemicellulose was made as a control group. These composite films were named as H30-PVA, H45-PVA, H60-PVA and H0-PVA, corresponding to corresponding to each component. H75 hemicellulose was not selected for the preparation of composite films due to its low yield and high lignin content.

### 3.5. Characteristics of Composite Film

Scanning electron microscopy (SEM, EVO MA15, Zeiss, Germany) was used to observe the morphology of the composite film. The thickness of the composite film was measured by a paper thickness gauge (Lorentzen and Wettre, Sweden). Each composite film was measured in three different areas and averaged with an accuracy of 1 µm. The composite film samples were cut into rectangles of 20 mm × 10 mm. The tensile stress of composite film was measured using a material tension tester (Instron Universal 5565, Canton, MA, USA) at room temperature with 50% humidity. The tensile speed was maintained at 5 mm/min. The tensile test was repeated three times and the average value was obtained. The UV shielding ability and visible light transmission ability of the composite film were measured by UV-Vis spectrophotometer (Cary 60, Agilent, Santa Clara, CA, USA) in the wavelength range of 900–190 nm. The thermal stability of the composite films was measured by thermogravimetric analyzer (TA Q500, Newcastle, DE, USA) under the following conditions: About 10 mg of film sample was heated from 60 °C to 650 °C at a constant heating rate of 10 °C/min. The nitrogen flow rate was 20 mL/min.

## 4. Conclusions

Loblolly pine hemicellulose with different structures was obtained by the alkali extraction-graded ethanol precipitation method, and the relationship between the physicochemical properties of the obtained hemicellulose and the properties of hemicellulose-PVA composite films were investigated. The results show that low concentration of ethanol is more likely to precipitate hemicelluloses with a longer straight chain and larger molecular weight, and a high concentration of ethanol was more likely to precipitate hemicellulose with a multi-branched, shorter chain and smaller molecular weights. Among all of the samples, H45-PVA has the best mechanical properties and H60-PVA shows the best UV-shielding ability. The thermal stability of the composite film is greatly influenced by the thermal stability of hemicellulose. The high molecular weight, high substitution and residual lignin content of hemicellulose can not only enhance its compatibility with the PVA composite film, but also improve the mechanical properties of the composite film. In addition, high substituted multi-branched hemicellulose is favored to increase the density and thermal stability of the film. The shielding ability of visible light will be better after adding hemicellulose with more side chains and high lignin content. This study provides a new perspective for future high value utilization of hemicellulose.

## Figures and Tables

**Figure 1 molecules-28-00046-f001:**
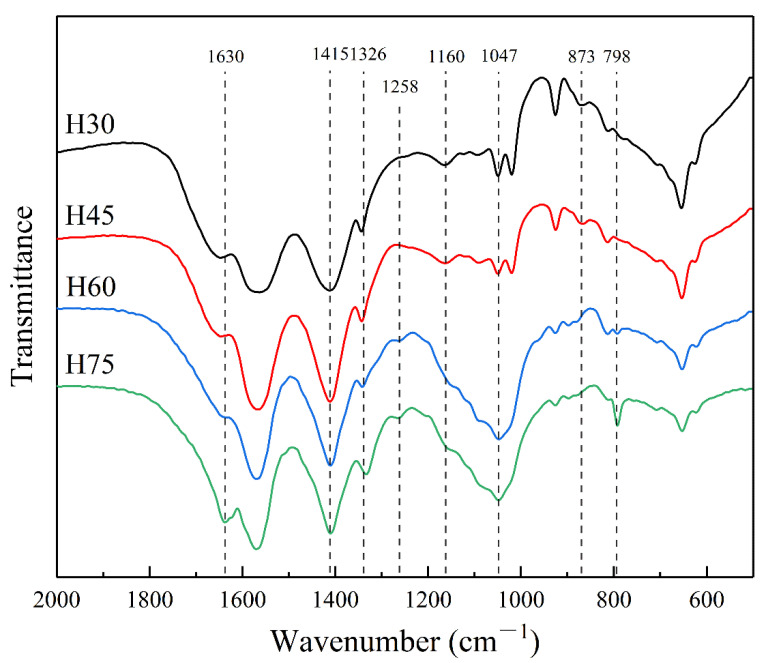
FT-IR spectra of graded ethanol precipitated hemicellulose.

**Figure 2 molecules-28-00046-f002:**
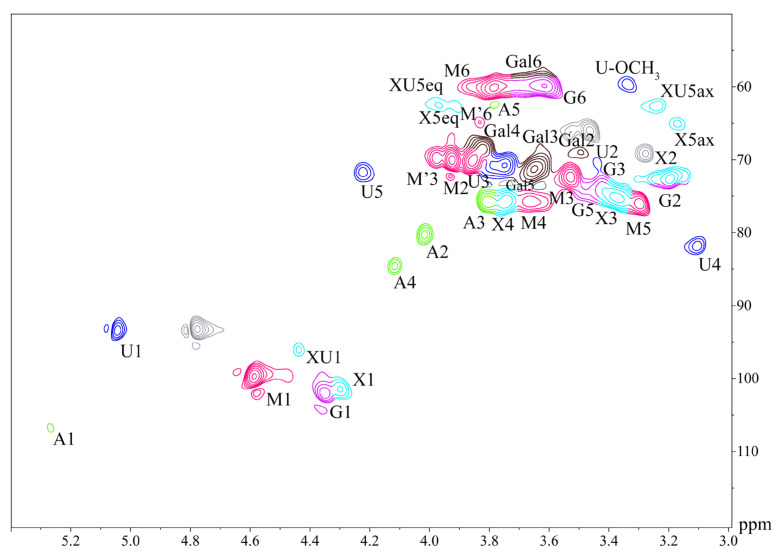
HSQC NMR spectrum of H45.

**Figure 3 molecules-28-00046-f003:**
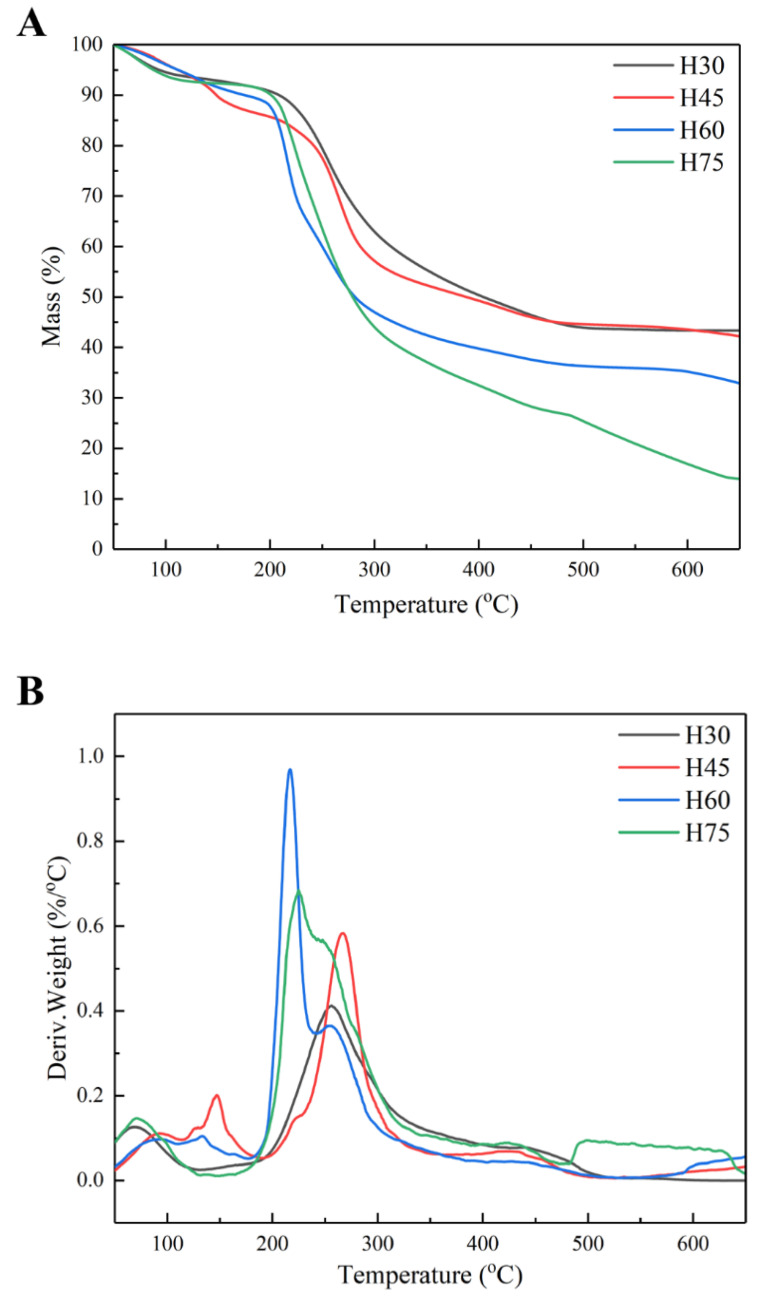
Thermogravimetric analysis of graded ethanol precipitated hemicellulose; (**A**): TG curve; (**B**): DTG curve.

**Figure 4 molecules-28-00046-f004:**
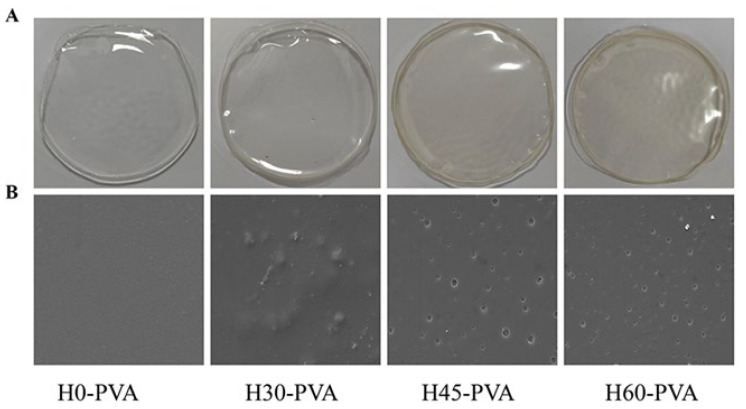
Laminated film picture, (**A**): composite film photo; (**B**): SEM image.

**Figure 5 molecules-28-00046-f005:**
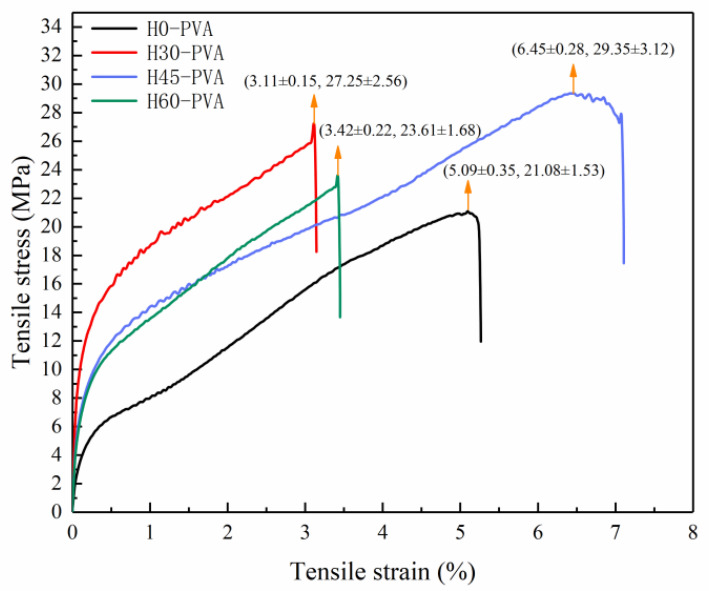
Stress-strain curve of hemicellulose-PVA composite film.

**Figure 6 molecules-28-00046-f006:**
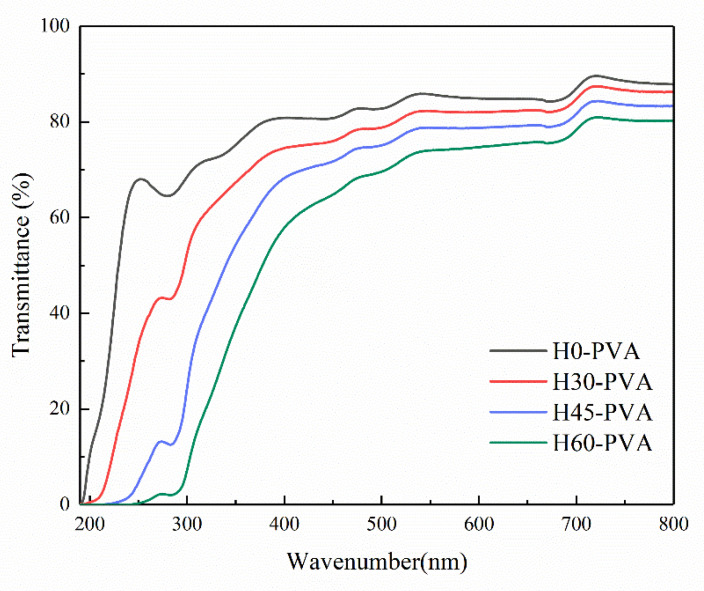
UV-Vis transmission curve of hemicellulose-PVA composite film.

**Figure 7 molecules-28-00046-f007:**
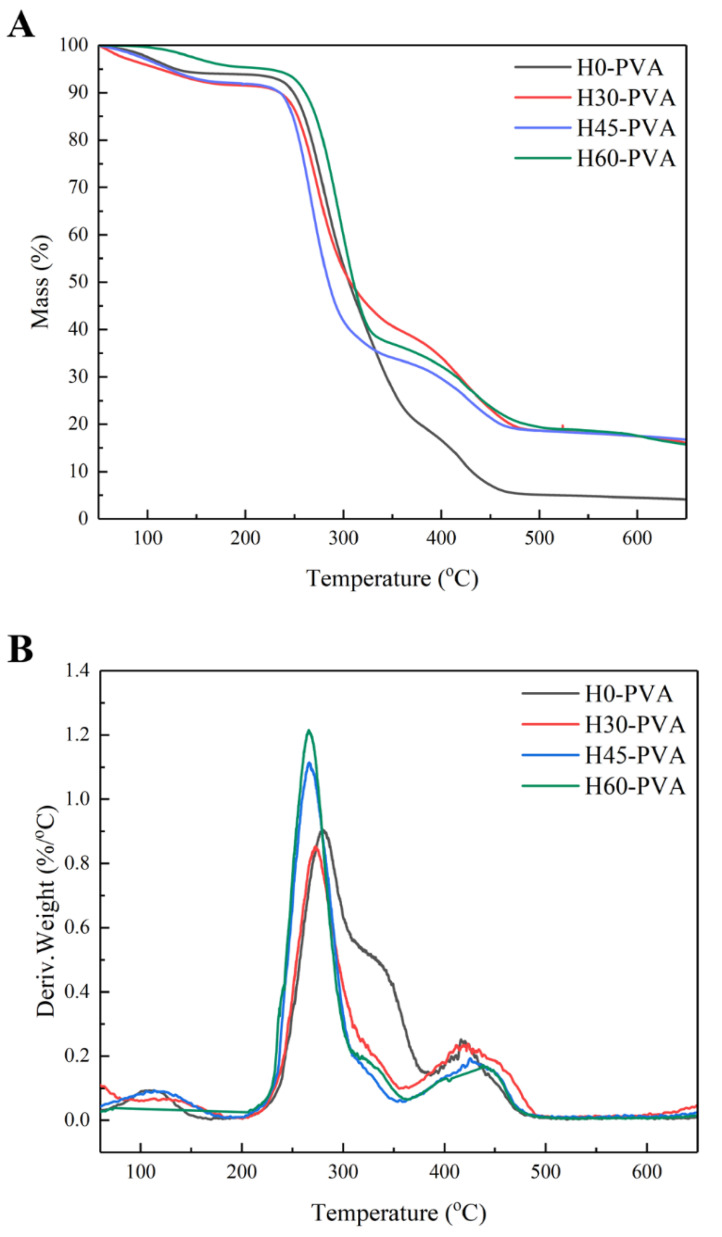
Thermogravimetric analysis of hemicellulose-PVA composite film; (**A**): TG curve; (**B**): DTG curve.

**Figure 8 molecules-28-00046-f008:**
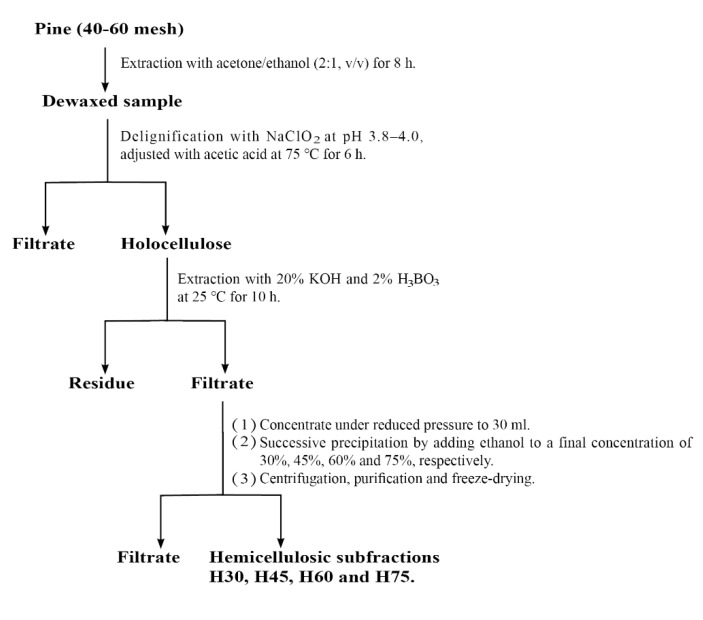
Scheme for the fractionation of alkali soluble hemicelluloses from loblolly pine by the graded ethanol precipitation.

**Table 1 molecules-28-00046-t001:** Yields of alkali-soluble hemicellulosic fractions precipitated with different concentrations of ethanol (weight percentage based on the pretreated material, %).

Fraction	H30	H45	H60	H75
Yield (%)	8.86 ± 0.68 ^a^	6.87 ± 0.24 ^b^	2.22 ± 0.42 ^c^	1.07 ± 0.12 ^d^

Data is presented as means ± SD (*n* = 4) and different letters indicate significant differences (*p* < 0.05) by Duncan’s multiple range test.

**Table 2 molecules-28-00046-t002:** Chemical compositions (expressed in relative weight percentage, %) of the alkali-soluble hemicellulosic fractions from loblolly pine.

Composition	H30	H45	H60	H75
Lignin	2.56 ± 0.13 ^d^	3.77 ± 0.22 ^c^	5.01 ± 0.43 ^b^	12.45 ± 1.60 ^a^
Arabinose	3.42 ± 0.02 ^a^	3.42 ± 0.32 ^a^	3.57 ± 0.20 ^a^	2.97 ± 0.31 ^b^
Galactose	7.66 ± 0.14 ^d^	10.08 ± 0.07 ^b^	11.02 ± 0.30 ^a^	8.58 ± 0.26 ^c^
Glucose	15.81 ± 0.15 ^a^	12.69 ± 0.14 ^b^	10.00 ± 0.21 ^c^	9.19 ± 0.09 ^d^
Xylose	23.97 ± 0.23 ^a^	21.03 ± 0.39 ^b^	17.10 ± 0.42 ^c^	14.9 ± 0.50 ^d^
Mannose	46.58 ± 0.47 ^d^	49.01 ± 0.05 ^c^	53.30 ± 0.57 ^a^	51.91 ± 0.48 ^b^
DS1	0.14	0.16	0.21	0.19
DS2	0.12	0.16	0.17	0.14

Note: DS1: xylan substitution, DS1 = arabinose content/xylose content; DS2: glucomannan substitution, DS2 = galactose content/(glucose content + mannose content), data is presented as means ± SD (*n* = 4) and different letters indicate significant differences (*p* < 0.05) by Duncan’s multiple range test.

**Table 3 molecules-28-00046-t003:** Weight-average (*Mw*) and number-average (*Mn*) molecular weights and polydispersity (*Mw*/*Mn*) of the hemicellulosic fractions.

Hemicellulosic Fractions	*Mw* (g/mol)	*Mn* (g/mol)	*Mw*/*Mn*
H30	71,295	42,072	1.69
H45	72,619	49,702	1.46
H60	53,771	33,675	1.60
H75	25,182	13,422	1.88

## Data Availability

The data presented in this study are available in the article.

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
