# Peer review of "Effect of Loblolly Pine (Pinus taeda L.) Hemicellulose Structure on the Properties of Hemicellulose-Polyvinyl Alcohol Composite Film"

_molecules, 2022, doi:10.3390/molecules28010046_

Round 1

Reviewer 1 Report

In this work the authors extracted various fractions of hemi-cellulose from a certain wood and performed an extensive characterization of these fractions. They also used these fractions as additives to PVA and they fabricated PVA-hemicellullose blend films. In general, this is a good work. There is a large portion of experimental results which are nicely analyzed and presented. However, there are four issues that must be addressed.

1.       Line 116. “low concentration of ethanol favored precipitation of xylan”.  In lines 109- 110 the authors mention that lignin content increases with increasing ethanol concentration. And in lines 112-133 they mention that lignin is mainly connected to xylan. The above three statements contradict each other. More precisely, if lignin is mainly bonded to xylan then both xylan and lignin should have the same dependency on ethanol’s concentration. I think it would be better to remove lines 110-116.

2.       I am not sure whether the authors have realized, but I think they have reached an interesting finding which is not adequately discussed and presented. This is the influence of lignin. For example, the authors in line 279 (and in the abstract) state that the better UV shielding activity is due to the presence of lignin. First of all, the authors should mention in the Introduction section whether the literature studies have taken into account the effect of lignin. For example, is the DP or Mw etc. of hemi-cellulose (of the literature and the one of the current study) influenced by lignin? If yes, then the conclusions about the molecular weight of hemi-cellulose and its influence on solubility, compatibility with PVA etc. could be revised by taking into account the effect of lignin. Lignin is a crosslinked polymer and thus may limit solubility in liquid solvent or mixing with other polymer. Perhaps the branches of hemi-cellulose in the high ethanol fractions are simply lignin which is also increased by increasing ethanol concentration? In few words the effect of lignin as an impurity may has been underestimated and it may deserve more extensive discussion (also in the Conclusion section there is no any reference to lignin).

3.       Typically, plasticizers are used at low concentrations to facilitate the processing of a polymer. In the introduction section (lines 66-67) the authors mention that for the preparation of hemi-cellulose films the addition of plasticizers is needed. Also in lines 79-80 the authors state that among the scope of their work is to use PVA as plasticizer for hemicellulose films. However, in lines 343-343 it is mentioned that 0.15 g of hemi-cellulose was mixed with 0.45g of PVA. So actually, hemi-cellulose can be considered as a plasticizer for PVA and not the opposite.

4.       English must also be improved. I would suggest extensive English editing because there are too may points that could be improved. However, I will suggest moderate English improvement because the errors are minor and the meaning is retained in the vast majority of cases. Below (suggestively) there are some points where English can be improved.

Lines 19 “…is favor for….”

Line 30 “…advantages of extensive, renewable….”

Line 88 “ hemicellulose….were…..”

Line 204 “It shows…”

Lines 207-208 “The third…..charring mass”

lines 311 “…take about….”

Author Response

请参阅附件。

Reviewer 2 Report

Abstract -> Please add rational of this research

L22 “basic” should be replaced with other word to show novelty/significance of this finding.

L84 Please recheck and confirm the format; sequence of Materials and methods and Results and discussion

Table 1, 2 Please recheck the statistical analysis as * is missing (there are only **).

L206 At high temperature much above 100 C there should be other volatization from other substances.

L231 Any precipitation of cellulose derivatives in water during preparation?

L239 Pore mostly came from bubbles in the solvent during preparation and pouring onto mold.

L265 Add more discussion e.g., Dispersion of cellulose possibly acted as reinforcement, improving strength of the polymer matrices which diversely reduced flexibility of the films (Phothisarattana et al., 2022: Colloids and Surfaces B: Biointerfaces).

Fig. 5 There should be average values and SD shown somewhere in the text.

L303 These natural based materials have high potential to serve as sustainable polymeric packaging which reduced environmental impact of petroleum-based plastic (Promhuad 2022 Polymers; Laorenza 2022 Polymers).

L303 What does “in nitrogen” mean?

Conclusions - > Some parts look like a repeat of Results. Please avoid repeating results and discussion. Key findings should be highly emphasized here.

Round 2

Reviewer 2 Report

The manuscript has been improved.